# Managing Reproduction in Hyperprolific Sow Herds

**DOI:** 10.3390/ani13111842

**Published:** 2023-06-01

**Authors:** Fernando Pandolfo Bortolozzo, Gabriela Piovesan Zanin, Rafael da Rosa Ulguim, Ana Paula Gonçalves Mellagi

**Affiliations:** Faculdade de Veterinária, Universidade Federal do Rio Grande do Sul (UFRGS), Setor de Suínos, Porto Alegre 90540-000, RS, Brazil

**Keywords:** gilt development, gestating sow, farrowing traits, preweaning mortality, surplus piglets, sow longevity, sow health

## Abstract

**Simple Summary:**

Understanding the physiological mechanisms responsible for the reproductive and productive development of modern sow genotypes will help to create or adapt management recommendations for the farm. This review describes the main challenges and management in the productive and reproductive phases of hyperprolific sows and some handling performed with the offspring. The described principles are critical points in the development of females, and understanding these principles better will lead to improved production rates and welfare in commercial pig production.

**Abstract:**

The rearing of large litters from hyperprolific sows is a characteristic of modern genotypes. However, these sows have body and reproductive characteristics that differentiate them from the genotypes of the past decades, making it necessary to adopt different management strategies. This review describes the main care and challenges associated with the hyperprolificity of sows during the period in which replacement gilts are selected, along with gestation, parturition, lactation, and the weaning-estrus interval. It describes the challenges that these sows’ piglets will face during the lactation period and includes some strategies adopted to develop these surplus piglets. In addition, it identifies areas where more research is needed to understand the reproductive management of modern genotypes.

## 1. Introduction

The modern sow genotype dramatically differs in terms of the productive and metabolic profile from genotypes of 20 to 30 years ago. Modern maternal lines have an impressive farrowing performance, with an increase of 4.5 total born piglets during the period 2006–2019 [1] or 6.7 total born during the period 2000–2020 [2]. One of the main drivers behind this continuous increase in litter size is genetic improvement, reaching farms through replacement gilts. Additionally, over the years, modern gilts have become leaner, more feed-efficient, and faster-growing than gilts of previous decades. Therefore, correct strategies in the selection, rearing, and puberty induction of replacement animals are important links for a constant flow of high-quality gilts into the breeding herd.

Despite efforts in genetic selection, hyperprolificity was accompanied by a decrease in piglet birth weight. As a result, piglets became more vulnerable, increasing pre-weaning mortality and compromising their potential growth. With larger litters, farrowing duration increased, resulting in stillbirths and dystocia, and it compromised the puerperal health of the sow. Furthermore, the amount of colostrum available per piglet has decreased and, in addition, the number of piglets born alive exceeds the number of functional teats. In this way, larger litters brought enormous challenges in the puerperal and lactation period. The lactational catabolism is intense in sows with more suckling piglets. Thus, there are great challenges in maintaining these females with adequate body conditions at weaning and throughout the gestation period. Additionally, piglets may experience more competition for udder access during the suckling period.

This review focuses on aspects of hyperprolific sows, highlighting the main changes in the reproductive physiology of these animals and providing an evidence-based approach for improved management interventions.

## 2. Birth Traits, Growth, and Body Development of Replacement Gilt

A constant flow of high-quality and well-managed gilts into the breeding herd has a significant impact on herd efficiency, improving longevity, stabilizing the structure of breeding herd parity, and allowing the farm to achieve its targets [3]. Successful selection for high prolificacy became associated with increases in the proportion of low-birth-weight pigs and preweaning mortality [4,5]. Additionally, the birth weight of the future breeding female directly affects her productivity and longevity in the herd. Gilts born with a weight less than 1 kg and selected for entry into the breeding herd show satisfactory results in terms of puberty onset and farrowing rates. However, the numbers of piglets produced over three parities and longevity were compromised [6]. More recently, when “litter of origin” was explored in the context of the increasing number of low-birth-weight pigs, a large percentage of low-birth-weight offspring came from a minority of sows with an extreme and repeatable low-litter-birth-weight phenotype (LLBWP) [7]. These sows probably have reduced placental development and capacity for efficient maternal–conceptus communication [8]. An LLBWP might carry all the same risks described above for individual low-birth-weight gilts but as a “litter” trait [9], and gilts born to a sow with an LLBWP have lower retention in the herd within 4 days of birth, at weaning, and at preselection into the breeding herd [10]. Considering the most extreme LLBWP population (bottom 15%), no sow first giving birth to a low-birth-weight litter produced a high-birth-weight litter at any subsequent farrowing [11]. Hence, the nucleus/multiplication farms can effectively select against extreme LLBWP sows without risking missing out on high-quality litters born in later parities [9].

Genetic selection and improvements in health, facilities, management, and nutrition have led to future breeders with exceptional weight gain and body development, particularly in the period between weaning and puberty induction [4]. The fast-growing modern genotype has less adipose tissue than its predecessors, modifying lifetime metabolic demands [1]. Typically, from weaning to puberty, gilts are fed ad libitum, and under production conditions, their nutrition is not always different from that of finishing pigs. Thus, contemporary gilts are fast-growing and have become a real challenge, particularly considering the proportion of gilts exceeding the target body weight at puberty and the negative impacts of being overweight on subsequent performance.

An adequate puberty stimulation process is essential to provide a consistent supply of eligible gilts for service [9]. The occurrence of puberty can be stimulated at younger ages by several environmental and management factors [12,13], but attempts to associate puberty with a critical weight [14,15], growth rate (GR) [16], fat [14,17], or lean tissue [17] are still controversial, being strongly influenced by the genotype.

The boar effect is a critical factor influencing puberty attainment in gilts, and daily exposure to a rotation of mature and high-libido boars maximizes the response [9,18]. Earlier data from Beltranena et al. [19] suggest that when lifetime GR is below 0.55 kg/day, puberty is delayed. More recent data support that with unrestricted feeding during the growing/finishing phase, GR is unlikely to limit the age of the first estrus [20,21]. Besides that, higher-lifetime-GR gilts (723 g/day) showed puberty earlier than lower-GR gilts (577 g/day), with more of them attaining puberty by 190 days of age (95% vs. 76%), showing the importance of lifetime GR as a factor anticipating puberty, mainly when gilts are exposed to boars at an early age [22]. A lifetime GR close to 800 g/d seems to have no detrimental effects on puberty onset. Even when the GR is higher than 620 g/d, puberty is not limited by GR [23].

Despite differences among the genetic lines, it is generally accepted that the target weight to breed gilts is 135 to 150 kg [3,9,24] or 140–160 kg and 13–15 mm backfat thickness at the age of 220–240 days [25]. With less attention paid to backfat thickness and age, gilts that reach this target weight with at least one previous recorded estrus and adapted to herd health status can be inseminated irrespective of age and backfat level [3,26]. Assuming a maternal weight gain of 35 to 40 kg during the first gestation, gilts would reach >180 kg after farrowing, which is protective against the detrimental effects of lean tissue loss during the first lactation [27].

There are no advantages in breeding overly heavy gilts since they will be overweight at farrowing, with more nutritional demands during lactation and over their productive life [3], compromising farrowing rate at the second parity [21]. Heavy gilts are also associated with locomotion problems over three parities and a higher occurrence of stillbirths [21,28], probably associated with greater difficulties in delivery. Therefore, from a practical standpoint, these animals should be bred earlier, respecting the target weight [26], or nutritional strategies could be used during the growing phase to reduce the GR of these gilts. These strategies may be associated with a reduction in the dietary Lys-to-energy ratio [29] or a 25% restriction in energy intake [30].

It is well accepted that maintaining a positive metabolic state in the pre-breeding period seems to be another critical step in optimizing herd reproductive performance [9]. However, the requirements of modern genotypes for a nutritional flushing strategy (increasing feed intake during the pre-breeding period) to increase ovulation rate are uncertain. More recently, it was observed that flush feeding after pubertal estrus is still important in improving the ovulation rate of modern replacement gilts [28]. The authors observed that flush feeding gilts during cycles 1 and 2 before breeding (2.1 vs. 3.6 kg/d) increased the BW and BF, whereas the ovulation rate at the third estrus was increased by increasing the feeding level in cycles 1 or 2. However, embryo survival was reduced when gilts were fed 3.6 kg/d during the cycle before breeding. In addition, it was observed that gilt weight influences litter size, and gilts with low backfat tended to respond more positively to a longer period with high pre-breeding feed allowance than fatter gilts [31]. Although more research is needed, the results available so far demonstrate that performing flush feeding during the first cycle after pubertal estrus brings an increase in the number of ovulations and the potential litter size, but it is probably not critical to those that are expected to reach or exceed the breeding target weight.

## 3. Controlling the Gestation Phase to Avoid Reproductive Failures

Conception failure and gestational loss are reported as the main culling reasons for sows [32,33,34]. Over the years, the advances in semen technologies and knowledge available in the literature on insemination techniques and female physiology for the definition of insemination protocols have minimized the impact of these procedures as the main reasons for reproductive failures [35]. However, housing type, environmental conditions, feeding system, and health status during the gestation phase are the most significant challenges nowadays.

Housing sows in collective pens was the main change adopted worldwide during gestation in the last 10 years. Optimum group size, moment of mixing, and feeding system are not precisely determined for collective pens. There is a consensus that the mixing of sows should not occur during the stress-sensitive stage of embryo implantation [36], which starts 11–12 days after insemination [37]. Differences in reproductive performance when mixing sows 4 weeks post-service were not observed compared to stall-housed sows [38,39]. However, conflicting responses were observed comparing sows mixed soon after insemination with those grouped 4 weeks post-insemination [36,38,39,40,41]. This difference among studies could partially relate to a variation in time in which the females were mixed. Reproductive performance was not affected in some studies mixing sows 4 days compared to 30 days post-insemination [36,40]. Nevertheless, when the mixing occurred between days 3 and 10 post-insemination compared with later mixing (≥30 d) or keeping in individual crates, decreases in conception and farrowing rates were observed [38,39]. The last-cited studies considered a large range between insemination and the day of mixing, which could impair reproductive performance. Thus, mixing close to the days of insemination does not seem to be detrimental in terms of gestational loss [42] when group housing soon after insemination is performed by farms.

The feeding system is probably the main challenge in defining the recommendation for group housing and mitigating the negative impacts on reproductive performance. Providing optimal conditions for individual feed access in collective pens is essential during the gestation phase since this is a period for adjustments in the body condition score (BCS), and different feeding programs are performed. The control of the BCS in collective pens is more difficult compared to individual crates, especially if adopting the mixing of females soon after insemination. The social ranking or feed space in group housing systems can result in females with restriction or overfeeding. In parity 1 and 2 sows housed in collective pens and mixed soon after insemination, a linear reduction in total piglets born was observed when an increase in daily feed intake (1.8, 2.5, or 3.2 kg/d) was provided from day 6 to 30 of gestation [28]. However, in a systematic review, Leal et al. [43] concluded that providing energy in excess of the maintenance requirement during early gestation did not impair reproductive performance, as suggested elsewhere [44]. Regardless of the conflicting responses, it is important to consider that most of the studies on this topic were performed in gilts [43] using no hyperprolific genotypes, recovering embryos, and using a reduced number of animals and levels of feed. In contrast, as reviewed elsewhere [44] feed restriction during the first 30 days of gestation resulted in a higher risk of losing the pregnancy. Thus, feed restriction in early gestation should be avoided. In young weaned sows, the recommended feeding strategy is to provide 1.5× the energy requirement for maintenance to prevent reproductive impairment [28]. Additional studies, including more feed levels in early gestation, should be considered for hyperprolific sows with evaluations up to and including farrowing performance.

Reproductive problems related to increased feed intake between days 30 to 90 of gestation are not a concern. However, the stress caused by disputes or feed restrictions is expected to compromise reproductive performance [45]. Some negative effects attributed to social stress on pregnancy maintenance can be related to nutritional issues [44]. Thus, increased feed consumption should be promoted between days 30 to 90 of gestation to avoid feed restriction and recover BCS. From day 90 of gestation, an increase in feed levels (bump-feeding) was extensively adopted for improving piglet birth weight, considering the higher growth of fetuses in this phase and the higher number of piglets in hyperprolific sows. However, studies using hyperprolific gilts and sows showed that the bump-feeding strategy was not effective in promoting an increase in piglet birth weight [46,47] and, in some cases, reduced voluntary feed intake during lactation [46,48]. The lack of improvements in piglet birth weight using bump-feeding could be attributed to the placental efficiency in transferring the additional nutrients for fetuses from day 90 of gestation. This hypothesis was evaluated by Mallmann et al. [28] by comparing the use of 1.8 vs. 3.5 kg/d between days 22 and 42 of gestation, followed by the use of 1.8 vs. 3.5 kg/d of feeding from day 90 until 110 of gestation in a factorial arrangement. The increase in feed intake at the beginning of gestation provided more nutrients for better placental development. However, total placental weight, placental efficiency, and birth weight of piglets were not affected by the different feeding levels or gestation phases. Other strategies for increasing piglet birth weight, including amino acids used in the third trimester of gestation, were reviewed by Moreira et al. [49] who indicated no beneficial effects regardless of the amino acid type. Based on the evidence in the literature, the inclusion of piglet birth weight in the genetic programs is the main strategy currently available to prevent low-birth-weight piglets in hyperprolific sows.

In the near future, more sows will be housed in collective pens worldwide. Comparisons among feed systems associated with other factors, such as density or group size in collective pens, and their impact on reproductive performance are complex, and respective studies are scarce. Thus, models for facilities for group housing are being developed by companies based on practical observations of welfare and reproductive performance. In our experience, the model that has been well adopted in Brazilian conditions considers the open short stalls for individual feeder access, a partially slated floor, a group size of 15–20 females with a minimum of 1.5 and 2 m^2^/animal for gestating gilts and sows, respectively.

The locomotor problems in group housing may increase, reducing reproductive performance and increasing culling for lameness [50]. A routine of daily individual care of animals is crucial for early treatment, avoiding gestation losses, welfare problems, and death of sows. Mortality of sows has been a concern in the last few years for multiple reasons [51]. Pelvic organ prolapse (POP) is one of the main reasons for death, with clinical signs occurring at the final gestation phase and structures prolapsed at farrowing and post-farrowing [52]. The total number of piglets born (indicating a high prolificacy) was evaluated as a risk factor for prolapses and was not associated with the problem [53]. The etiology of POP is not fully determined, but a low BCS was reported as an important risk factor [54]. Additionally, farms that use an increased feed intake during late gestation for sows having a low condition score had lower POP rates compared to farms not using a bump-feeding strategy. Thus, correct adjustment of the BCS during the gestation phase is necessary to avoid POP and reduce mortality, since farms with reduced farrowing rates are associated with high sow mortality [55].

## 4. Farrowing Traits of Hyperprolific Sows and Their Consequences

One of the benchmarks used worldwide is the weaned pig output, and it is noticeable that the improvements made by the genetic companies in prolificity are observed in commercial herds. However, piglet loss rates before weaning are still high (Figure 1), despite all efforts accomplished in facilities, management, and new traits for the selection program. Generally, approximately 15.0% of piglets do not survive from birth to weaning [56]. High piglet mortality can be associated with farrowing events, such as prolonged farrowings, dystocia, and hypoxia [57].

The duration of farrowing has been increasing over the years, presumably due to the increase in litter size. For example, in 2005, the average farrowing duration was 166 min [58], which has now increased to 268 min [59]. Several factors can affect the duration of farrowing, such as obesity, constipation, housing system [60], litter size, number of stillbirths [58,60], the use of oxytocin [61], and genetics [58]. The consequences of prolonged farrowing and dystocia are, in most cases, negative to the sow [62]. The uterine environment is exposed to contamination, and pathogenic agents are easily carried into its interior when there are obstetric manipulations. Hyperthermia in the first 3 days postpartum is common, as well as a decrease in appetite and water consumption [63], compromising the general health of the sows [64] and predisposing them to a reduction in milk production and subsequent performance [65,66].

## 5. Neonatal Losses: Incidence of Stillbirths and Low Viability Piglets

Prolonged farrowing is strongly correlated with litter size and stillbirths. Indeed, sows with farrowing >3 h had 2.0 times higher odds of stillbirth occurrence than sows with shorter farrowing [67]. Oliviero et al. [60] reported that a long farrowing duration (>300 min) resulted in an average of 1.5 stillborn piglets per litter, whereas farrowing of normal duration (<300 min) resulted in 0.4 stillborn piglets. Similarly, females with 0 and 2 stillbirths had farrowing lengths of 307 and 487 min, respectively [59]. However, it is unclear whether the presence of stillbirths is the cause or the consequence of prolonged parturition.

The consequences of fetal distress, whether caused by prolonged farrowing or dystocia, imply lower piglet viability after birth [68], with reports of 7.9 times more chances of death until the third day of life in piglets born cyanotic [69]. The last piglets born are more prone to stillbirth, with reports of stillbirth rates of 3.6%, 10.1%, and 21.7% in piglets from the 1st to 9th, 10th to 13th, and 14th order, respectively [67]. It has also been reported that from the 11th piglet born, the chance of being stillborn is approximately 15 times greater than that for piglets with a birth order below 7 [70]. Piglets that take longer to be born tend to have the umbilical cord broken at birth [71], and consequently, their chances of death by the third day may increase 3-fold when compared to piglets with an intact cord [69]. Thus, it is clear that the factors influencing stillbirth also impact the occurrence of low-viability piglets.

## 6. Colostrum Yield

Hyperprolific herds may have to deal with reduced individual colostrum intake by newborn piglets because it is expected that as litter size increases, piglet colostrum intake decreases [72]. Indeed, Theil et al. [25] noticed an increase in colostrum yield over 13 years (5.3 to 6.9 kg per sow), but with a concomitant reduction in colostrum intake of individual piglets (438 to 322 g/piglet), and as previous reports have shown no association between litter size and colostrum yield [73,74], the reduction in individual colostrum intake in large litters may depend on the presence of low-birth-weight piglets and increased teat disputes. Additionally, piglets with low vitality may delay consuming colostrum, increasing the risk of death. Thus, it is paramount that some practices must be adapted in farrowing supervision and day-1 piglet care protocols to increase udder access and colostrum consumption.

Although farrowing duration does not affect colostrum production [75,76], females with low colostrum production (<3 kg) had more stillbirths and longer intervals between piglet births in the early parturition phase [75]. This effect can be attributed, at least in part, to the maternal endocrine status, since many hormones have important role in lactogenesis and parturition [73], indicating that these events are linked.

## 7. Female Health, Reproductive Performance, and Longevity

Farrowing is physically demanding and sows become prone to exhaustion, heart failure, physical injury, and reduced immunity [77]. The duration of parturition can influence general and uterine health. Long parturitions may increase the percentage of females with hyperthermia (≥39.0 °C) in the first 24 h postpartum and reduce their appetite [63]. One of the clinical conditions frequently observed is post-partum dysgalactia syndrome (PPDS), characterized by insufficient colostrum and milk production in the first days postpartum [78]. Additionally, constipation on the day of parturition reduces the female appetite, increases the occurrence of PPDS, and enhances the percentage of females with hyperthermia [79]. In addition, the accumulation of fluid in the uterus in the postpartum period may predispose to puerperal metritis [59]. Although the disease is, in most cases, temporary and reversible, the piglets will be affected, especially those litters that had a low birth weight, due to the compromised milk production in the first days of life [80]. Ultrasound images are interesting tools to evaluate female reproductive soundness, detecting the retention of placenta or endometritis [64]. Postpartum ultrasound can be used to identify uterus enlargement and fluid in females having manual obstetric intervention and prolonged farrowing [59]. Additionally, the visualization of piglets might indicate obstruction or uterine inertia [64].

Prolonged farrowings and dystocia negatively influence uterine health and increase the risk of reproductive failure. Uterine involution can be delayed by high BCS values and hyperthermia [81], prolonged parturition, and delayed placenta expulsion [59]. Regarding the course of parturition, females with a parturition duration ≥300 min were 1.5 time more likely to have weaning-to-estrus intervals prolonged by 1 day and have an odds ratio of 3.4 to return to estrus after the first mating compared to sows that had a farrowing interval <300 min [65]. Sow longevity can be compromised by dystocia or lack of correct assistance. A prolonged weaning-to-estrus interval, a reduced farrowing rate, a small litter size in the subsequent reproductive cycle, and a greater culling rate were reported in different parity sows with manual obstetric intervention [82]. Manual obstetric intervention should be performed prudently to avoid the risk of postpartum metritis and delayed uterine involution, compromising the subsequent reproductive performance and longevity of the sow.

The peripartum period is critical for sow death due to heart failure, urogenital disorders, and prolapses [52,83,84,85]. Sows in confinement systems are particularly prone to heat stress [77]. In an individual crate system, Vearick et al. [85] observed a tendency for higher mortality when the ambient temperature was above 33 °C. Similarly, in hot and humid seasons, a higher occurrence of death was observed, especially for young females [83]. Thus, management to reduce heat stress is essential, mainly in sows with a high body condition score. Heat stress may play an important role in physical fatigue of hyperprolific sows, and appropriate care during the peripartum period, by ensuring sow comfort and water intake, may alleviate distress and reduce sow mortality.

In recent years, an important increase in the incidence of POP (including uterus, vagina and/or rectum) has been documented worldwide, with increased sow mortality and culling rates. Genetic involvement was not well established and since there is a low heritability [86], prolapse incidence is probably related to environmental and management effects. Although high litter size may exert more pressure on the uterus and abdomen, a survey representing almost 400,000 sows found no evidence of an effect of litter size on POP occurrence [87]. Contrarily, Iida et al. [53] reported that prolapse incidence was higher in sows with a small litter size, the presence of two or more stillborn piglets, and parity greater than 3. Usually, a small litter size is related to heavier piglets, which can also compromise the birth canal. Farrowing assistance and farrowing induction were not associated with POP occurrence in that study. Other factors contributing to POP may include zearalenone, high-density diets, vitamin deficiency, lack of adequate water, and excessive slope of the floor [88]. Currently, although the etiology is not fully understood, it is possible to identify low- and high-risk females according to perineal scores based on swelling, redness, and protrusion, varying from 1 to 3 [89]. Score 3 sows represent a low proportion, reaching 3–5% of the population, but more than 20% of them experienced POP [90]. Additionally, some blood markers were identified as putative for score 3 sows, such as higher serum androsterone, estradiol, and lipopolysaccharide-binding protein levels and decreased numbers of circulating lymphocytes and monocytes.

## 8. Strategies and Tools to Optimize Performance at Farrowing

Reducing farrowing duration involves several approaches. Although uterotonics (e.g., oxytocin or its analogs) can reduce farrowing duration, administration in the early process increases the stillbirth rate [61,91,92]. Recently, it has been shown that carbetocin is also associated with reduced colostrum yield [91,93]. In general, however, the studies conducted with uterotonics did not consider individual necessity and used different dosages and timing. It is established that uterotonics should be administered if there is no obstruction, in the presence of weak contractions and after other obstetric assistance measures. Manual obstetric intervention is another practice to assist when the birth interval of piglets is increased, aiming to increase piglet survival and reduce farrowing duration. This procedure is more frequently observed in older parity sows and during summer [82], probably due to the lower muscle tone and physical exhaustion.

In many farms, sows are fed once a day until farrowing, resulting in a prolonged period of low energy availability. As parturition is an energy-demanding event for the sow, expelling a large litter size is challenging. Additionally, constipation a few days before parturition is expected because of the low feed consumption by the female [94], which may result in prolonged farrowing [60]. Prepartum feed restriction leads to hypoglycemia and less energy available for the intense contraction that the uterus and abdominal muscles will require [95]. Appropriate feeding of sows in the transition phase from the end of gestation to early lactation is essential to assure suitable fetal and mammary growth, farrowing, and colostrum production [25]. Increasing sow fiber intake prior to parturition can alleviate constipation degree [96]. The effect on the farrowing process and newborn piglet survival is still inconsistent. Whilst some studies have reported no effect of fiber on farrowing length [97,98], others have shown promising findings on the reduction of the expulsion phase and/or stillbirth rate [99,100]. Some possible reasons could be related to the fiber substrates and levels since fiber supplementation higher than 20% seemed to lead to better results [99].

Feyera et al. [99] demonstrated a negative correlation between blood glucose levels and farrowing length. In that study, the interval from the last meal until the onset of farrowing affected the farrowing duration, suggesting that sows in late gestation should receive at least three meals daily to avoid low glucose concentrations at farrowing. Glucocorticoids can increase glucose production by stimulating proteolysis, lipolysis, and hepatic gluconeogenesis [101]. Recently, it has been reported that dexamethasone treatment (20 mg) before farrowing onset (at the time of copious colostrum secretion) can increase the glucose concentration throughout the farrowing process, reduce farrowing duration, and decrease the necessity for obstetric interventions for primiparous sows [102]. Although dexamethasone treatment resulted in a lower occurrence of piglets born with meconium staining, no effects on stillbirth rates, piglet survival, or performance until 5 d of age were found [102]. Contrarily, treatment with 20 mg dexamethasone 24 h after farrowing induction (at 113 days of gestation) did not influence the duration of farrowing, the incidence of dystocia, or preweaning survival [103]. Further investigations may help to establish a strategic protocol for treating sows more prone to dystocia and stillbirths and to determine the optimal moment for dexamethasone treatment.

At the herd level, controlling sow weight and body condition is essential to obtain better performance, herd longevity, and optimal feed use. Females with high backfat (>17 mm) have shown 155 min-longer farrowings [60] compared to low backfat thickness. Delayed progesterone decline at peripartum and more fat around the birth canal are believed to be involved in this phenomenon [94]. It is important to keep in mind that overweight sows may have reduced colostrum production [104] and limited feed intake during lactation, compromising milk production [105,106]. Maternal undernutrition also compromises sow and offspring performance, with increases observed in the stillbirth rate and colostrum/milk production [106]. Decaluwé et al. [107] showed a negative association between colostrum yield and backfat loss during the last week of gestation, indicating that body mobilization cannot fully compensate for undernutrition during mammogenesis.

## 9. Managing Lactating and Weaned Sows

Milk production for large litters requires a significant body reserve mobilization as nutrient intake is prioritized for colostrum and milk production. Feed intake, however, is usually insufficient to meet all nutritional requirements and sows mobilize body reserves, resulting in a catabolic state [105]. In addition to milk production, primiparous sows have not reached adult body weight, which deserves greater attention during the lactation period [108].

In the past, catabolic sows were expected to have longer weaning-to-estrus intervals (WEI), lower ovulation rates, low embryo survival, and smaller litter size [109]. In the past two decades, there has been evidence that modern genotypes are more resilient on return to estrus after weaning, but the negative effects on conception rate and subsequent litter size persist [110,111,112,113,114]. Follicle development may be compromised under a catabolic sow state, impairing the ovulation rate and embryo survival.

The modern genotypes manifest estrus in an average of 4.8 days after weaning [115] and have a lower anestrus rate [116]. By 1 week after weaning, 90–95% of the multiparous sows are expected to exhibit estrus [115]. Thus, recovering body reserves before breeding is difficult for sows. Whilst in the past, ad libitum access to feed was a common strategy for sows with excessive loss of body reserves, recent findings suggest that increased feed intake during WEI reduces body reserve loss during WEI, but with no improvement in follicular growth and subsequent reproductive performance [116]. Thus, minimizing maternal body loss is still critical for modern hyperprolific genotypes. Strategies to reduce excessive weight loss include nutritional gilt and gestating female management, as previously discussed, controlling temperatures in farrowing units, and ad libitum access to feed and water and health care for lactating sows.

In recent years, the swine industry has focused on increasing pig weaning age. Although many advantages are reported for piglets, as further discussed, increasing lactation length may be associated with different management strategies for sows. A longer lactation period facilitates better uterine involution before insemination. For each day that the farrowing–weaning interval increased, the sows produced 0.19 more piglets per year or 0.32 more piglets until the fourth farrowing [117]. In some countries where lactation may take longer than 5 weeks, an improvement in the subsequent litter size is reported due to the restored body condition during lactation [114].

## 10. Management of Hyperprolific Litters in the Lactational Period

With the increasing number of large litters, the sow needs to produce more milk, but the amount of milk available for each piglet decreases due to increased competition for teats. Insufficient colostrum and milk intake might cause problems during the pre- and post-weaning phases. Pre-weaning mortality is a multifactorial process resulting from interactions among the sow, the piglet, and the environment [118]. Although this is a challenge for large litters, decreasing pre-weaning mortality is an excellent opportunity to enhance animal welfare and to make pig production more sustainable.

### 10.1. Number of Piglets and Functional Teats

A functional teat must provide enough milk to develop a piglet [119]. Surplus piglets result in delayed colostrum intake, higher mortality rates, more suckling failures, reduced litter growth, or more face lesions due to increased competition for teats [120]. In fact, sows with more functional teats have larger litters and piglets at weaning compared to sows with fewer functional teats [121]. In one study, an increase of one available teat improved survival by 3.25%, the weight of litters at weaning tended to increase by 3.6 kg, the number of weaned piglets increased by 0.34 piglets [122], and pre-weaning mortality was reduced by 3.04% [123].

Sows do not have a mammary cistern, and milk is available to the piglets only in short intervals of time, reflecting the need for a functional teat for each piglet. With a moderate heritability for a functional teat (h^2^ = 0.31) [124], the importance of including functional teats in genetic improvement programs to increase the survival of piglets and contribute to the development of litters is evident. Thus, strategies must be addressed to develop the piglets exceeding the number of functional teats and will be discussed in this review.

### 10.2. Cross-Fostering Protocols

Cross-fostering is a strategy used to reduce weight variation within the litter or to keep a number of piglets that can consume colostrum and milk adequately, removing the piglets surplus to the number of functional teats. It is usually performed during the first 2 days of life before teat order is established and colostrum consumption is ensured. However, there is no standard protocol among the farms. The litter can be composed of biological, adoptive, or mixed piglets, with a survival rate until weaning of approximately 97.2% [125].

Nevertheless, care must be taken in how the cross-fostering of the piglets is carried out. Souza et al. [126] observed that light piglets should not be cross-fostered with heavy piglets to avoid losing suckling episodes and compromised survival rate. Similarly, Vande Pol et al. [127] highlighted the recommendation of cross-fostering light piglets in litters with uniform weight. However, no difference was observed when light piglets were cross-fostered with intermediate-weight piglets [128].

The number of functional teats in the modern female is, on average, 13.9 [129], and the number of piglets born alive is above 15. Thus, the number of teats is a limiting factor for the development of modern large litters and should be addressed in cross-fostering protocols. Recently, Vande Pol et al. [130] used the number of functional teats to develop cross-fostering protocols and observed that the group with two piglets exceeding the number of functional teats had higher pre-weaning mortality, even though these sows weaned numerically more piglets (+0.7 piglets) and had a similar weaning weight compared to the group cross-fostering with the same number of piglets and teats. As pre-weaning mortality is concentrated during the first days of life, females are prone to have more non-stimulated teats during lactation. In practice, some companies have performed cross-fostering with more piglets than functional teats, with few scientific findings. Thus, further studies are needed to evaluate how many piglets per teat can be fostered, how different categories of piglets (light/heavy) will respond to this management, and how long after farrowing fostering can be successfully performed. Establishing a standardized protocol will assist producers in forming litters with a better ability to grow and survive.

### 10.3. Pre-Weaning Litter Socialization

An alternative to improve pre-and post-weaning performance is the socialization of the litter. Research that removed the barrier between two adjacent pens, allowing piglets to interact before weaning, showed a reduction in aggressive behavior and the occurrence of injuries [131,132,133]. In addition, a better growth rate was reported in the post-weaning phase [131,132]. In traditional farrowing crate systems, performance failures in piglets are common due to disputes for teats or low milk production by a mammary gland, which may inhibit adequate piglet growth. By allowing access to more than one sow, piglets have the opportunity to access other teats (cross-suckling) that could otherwise be available and unused.

In a previous study, however, a greater score of teat lesions was also found in sows, along with skin lesions in piglets from socialized litters (performed on day 14 of lactation). Interestingly, socialized pigs had 19% fewer skin lesions than the control group after weaning [134]. Irrespective of the time of socialization (7 or 14 days of life), piglets showed more aggressive and play behaviors than control piglets; however, the number of skin lesions indicated that co-mingled piglets fought less [133].

Other studies propose a system with more physical space and social enrichment for multi-suckling sows and piglets [135]. Post-weaning benefits associated with multi-suckling systems include higher feed intake and weight gain and a lower occurrence of tail biting [135]. Thus, litter socialization, by removing the barrier between pens or by multi-suckling, is an area that requires further research to assess the benefits and challenges for sow and litter welfare.

### 10.4. Nurse Sows and Supplementary Feeding

In many hyperprolific herds, nurse sows may represent 10–15% of lactating females, with some herds showing a prevalence as high as 50% [57]. Two strategies are currently used. In the one-step strategy, a recently weaned sow raises surplus newborn piglets. In the two-step strategy, the weaned sow raises a litter from an intermediate sow after 5–10 days of lactation. The intermediate sow is moved to foster the newborn surplus piglets. In both strategies, the lactation length is prolonged, with the aim to increase the pre-weaning survival of piglets in a given farrowing group. Although a common practice among farms, there is a great variation in the methodology, making difficult to stablish a standard protocol [57].

Further, some sows may not accept the foster litter or may experience reduced milk production due to latency to first nursing. Foster piglets present more disputes for teats, missed nursing, and lower growth rates [136]. Prolonged lactation may result in teat injuries, poor body condition at weaning, and locomotor problems. Another implication is the dissemination of pathogenic agents. According to Garrido-Mantilla et al. [137], this management strategy can facilitate the transmission of the Influenza A and PRRS viruses between nursing sows and piglets.

Nutritional supplementation may be an alternative to increasing the vitality of the piglet and its ability to ingest colostrum/milk, helping to compensate for low body reserves. Piglets consuming low amounts of colostrum had increased survival rates when receiving an energy supplement (from 66.7% to 93.3%) [138]. Additionally, providing oral supplements may improve preweaning survival [139] and the weaning weight of low-birth-weight piglets [140].

Artificial rearing involves keeping piglets in specialized enclosures equipped with lamps, water and milk replacer cups, and solid creep feed. This system can substitute nurse sows or may be used to raise sick and starving piglets. As piglets have free access to milk replacer, weaning weight is higher, but welfare and post-weaning performance are occasionally negatively influenced [57].

Alternatively, milk replacers can be offered in cups, installed in the farrowing pens, and activated by the piglet snout. This management offers interesting results in large litters, minimizing piglet mortality, improving weaning weight, and decreasing within-litter variation [141]. The major limitations of supplementation for piglets, as the sole or complementary nutritional source, are the high investment and maintenance of these systems, in addition to problems related to animal welfare when piglets are deprived of maternal contact.

### 10.5. Increased Weaning Age

The weaning period represents an abrupt transition from liquid to solid feed and the formation of a new social hierarchy (housing with “unknown” piglets). Despite the provision of highly digestible diets after weaning, this practice is associated with an abrupt reduction in feed consumption in the first days, combined with hierarchical stress generated in the new facility [142].

Later weaning can benefit the development of the intestinal barrier and reduce immunological parameters linked to stress [143]. Increasing the weaning age from 19 to 28 days positively affected pig performance in the nursery and increased body weight sold per pig weaned, indicating 25 days as the optimal age [144]. Furthermore, increasing the weaning age from 18.5 to 24.5 days positively affected pig performance by reducing the number of pigs treated, increasing the average daily gain and feed intake, and marginally reducing total pig losses [145]. Thus, increasing weaning age is particularly important in systems where antibiotic use is more restricted.

## 11. Conclusions

Hyperprolific herds are a challenge for pig farming since females present distinct requirements, and piglets may struggle to develop and survive. As increasing the number of functional teats may take several years, some of the issues discussed in this review will persist. Alongside this, the growing concern for animal welfare, such as group-housing gestation and increased weaning age, requires more efforts to ensure that higher prolificity results in a better quantity and quality of weaned piglets, along with improving sow longevity. Consequently, the management of hyperprolific herds includes well-managed replacement gilts, enhanced management during pregnancy, attention to prolonged farrowing, practices aiming to increase the consumption of colostrum and milk, and strategies to raise piglets exceeding the number of functional teats available in the farrowing group. It is evident, thus, that overall pig management must be improved to meet sow, piglet, and industry demands. Although we highlighted important sow and litter handling to maximize pig production, the magnitude of each influencing factor may differ among the farms. Thus, it is paramount that each system identifies all the aspects previously discussed to implement the recommended management.

## Figures and Tables

**Figure 1 animals-13-01842-f001:**
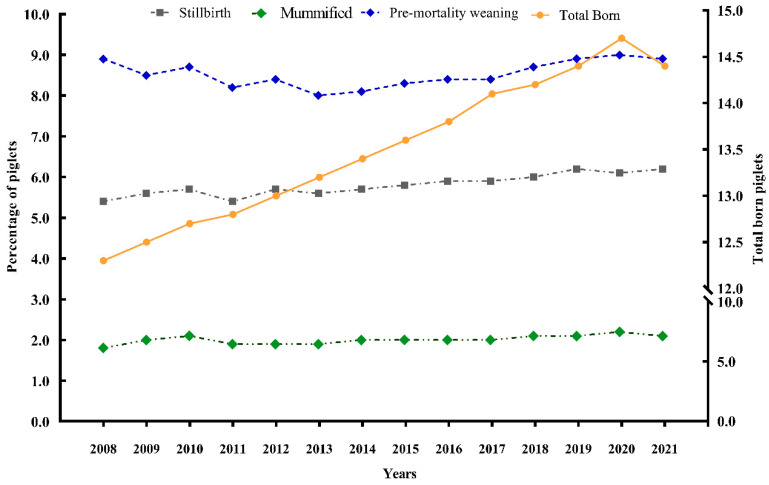
Brazilian trends in number of piglet total born, percentage of mummies, stillborn, pre-weaning mortality. Data from 2021 collected from 1715 herds and 1,538,548 sows [56].

## Data Availability

Data supporting reported results can be sent to anyone interested by contacting the corresponding author.

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
