# Peer review of "Managing Reproduction in Hyperprolific Sow Herds"

_animals, 2023, doi:10.3390/ani13111842_

Round 1

Reviewer 1 Report

Overall, this is a good thorough review of the literature on this topic. 

Please see the attached file comments for suggestions on improvements in wording for clarity and content as well as areas where citations may be needed. 

Author Response

Dear reviewer,

First of all, we would like to thank you for the helpful suggestions and comments about our manuscript (ANIMALS-2339161 “Managing Reproduction in (Hyper) Prolific Sows”). A file with a Revision Note (together with the Manuscript), in which are described the changes performed according to the suggestions/questions made by the reviewers/editors, was submitted to the editors. The grammar and the suggestions/questions have been corrected (and highlighted) direct in the manuscript.

Best regards

Fernando Bortolozzo 

Reviewer 2 Report

The review article entitled “Controlling Reproduction in (Hyper) Prolific Sows by Fernando PandolfoBortolozzo; Gabriela PiovesanZanin; Rafael da Rosa Ulguim; Ana Paula GonçalvesMellagi describes the issues of hyerprolific sows fertility and strategies to improve the fertility. However, the manuscript lack these things and even the write-up is so strange that it is very difficult to understand.

The sentence structure is very poor and there are major grammatical mistakes in manuscript.

The review objectives are not well defined and even the study questions are lacking

Although the author mentioned repeatedly about the genetic selection but did not focus on this aspect.

There is no synchrony between headings and paragraphs and all irregular and irrelevant information is incorporated.

Thus at my end, it is difficult to accept such reviews for publication in Animals and authors are encouraged to polish the manuscript, create novelty and readability in the manuscript.

Author Response

Dear reviewer,

First of all, we would like to thank you for the helpful suggestions and comments about our manuscript (ANIMALS-2339161 “Managing Reproduction in (Hyper) Prolific Sows”). A file with a Revision Note (together with the Manuscript), in which are described the changes performed according to the suggestions/questions made by the reviewers/editors, was submitted to the editors. The grammar and the suggestions/questions have been corrected (and highlighted) direct in the manuscript.

Dear reviewer, we would like to thank you for your time and comments regarding our manuscript. After all genetic efforts, modern sows became hyperprolific. Although the improvement made in litter size, other aspects are limiting or challenging, as we discussed in this review. Thus, the text is focused on the management of hyperprolific sows, not on genetic programs. The manuscript has been professionally proofread before submission. We have considered all suggestions made by the editors and reviewer 1 on grammar matter.

Best regards,

Fernando Bortolozzo 

Reviewer 3 Report

Hyperprolific herds are a challenge for pig farming. This review systematically analyzes the factors that influence the reproductive efficiency of modern breeding prolific sows in terms of gilt management, pregnancy management, boars, piglets, and nutrition management.

I have two doubts:

1. The manuscript systematically analyzes all aspects that affect the reproductive performance of prolific sows. Can the author intuitively rank the influencing factors according to their magnitude in the form of a table?

2. Does the author get a set of effective management strategies?

Author Response

Dear reviewer,

First of all, we would like to thank you for the helpful suggestions and comments about our manuscript (ANIMALS-2339161 “Managing Reproduction in (Hyper) Prolific Sows”). A file with a Revision Note (together with the Manuscript), in which are described the changes performed according to the suggestions/questions made by the reviewers/editors, was submitted to the editors. The grammar and the suggestions/questions have been corrected (and highlighted) direct in the manuscript.

Thank you for your suggestion. In our opinion, a rank of factors according to their importance would differ among herds. Besides, due to the length of the manuscript, we preferred not to include a table. However, in the conclusion section, we add a comment regarding your concern (now Lines 562-565).

Best regards,

Fernando Bortolozzo 

Round 2

Reviewer 2 Report

Although this review is focusing on the management of reproduction in (hyper) prolific sows but they didn't discuss about the anestrous either due to season or ovarian dysfunction.

Further, the reproductive management necessitate utilization of estrous synchronization protocols for having litter in favorable seasons to reduce the piglet’s mortality due to harsh season (cold or hot).

Nutrition is another important aspect and they just mentioned about nutritional deficiency rather did not discuss about nutritional plans for improving fertility. They just focused on few aspects of managing reproduction.

Actually, the topic discussed is all about managemental issues not fully focing reproduction as they did not discuss about infertility disorders so the authors must consider title revision as per their contents.

Author Response

Dear reviewer, we apologize for not responding appropriately to your opinion. A point-by-point response is  attached as a pdf file.
